# Robot-assisted gait training in patients with various neurological diseases: A mixed methods feasibility study

Isabella Hotz[1], Sarah Mildner[1], Michaela Stampfer-Kountchev[2], Bianca Slamik[2], Christoph Blättner[2], Elisabeth Türtscher[2], Franziska Kübler[1], Clemens Höfer[2], Johanna Panzl[2], Michael Rücker[2], Christian Brenneis[2,3], Barbara Seebacher[1,3,4]*

1 Department of Rehabilitation Science, Clinic for Rehabilitation Münster, Münster, Austria, 2 Department of Neurology, Clinic for Rehabilitation Münster, Münster, Austria, 3 Karl Landsteiner Institute of Interdisciplinary Rehabilitation Research, Münster, Austria, 4 Clinical Department of Neurology, Medical University of Innsbruck, Innsbruck, Austria

* barbara.seebacher@reha-muenster.at

## Abstract

### Background

Walking impairment represents a relevant symptom in patients with neurological diseases often compromising social participation. Currently, mixed methods studies on robot-assisted gait training (RAGT) in patients with rare neurological diseases are lacking. This study aimed to explore the feasibility, acceptability, goal attainment and preliminary effects of RAGT in patients with common and rare neurological diseases and understand the intervention context and process.

### Methods

A mixed-methods feasibility study was conducted at an Austrian rehabilitation centre. Twenty-eight inpatients after stroke in the subacute and chronic phases, with multiple sclerosis, Parkinson's disease, spinal cord injury, spinocerebellar ataxia, acute/chronic inflammatory demyelinating polyneuropathy and motor neuron disease were included. Patients received RAGT for 45 minutes, 4x/week, for 4 weeks. Baseline and post-intervention assessments included gait parameters, walking and balance, and questionnaires. Semi-structured observations were conducted twice during the intervention period and analysed using thematic analysis. Descriptive statistics within the respective disease groups and calculation of effect sizes for the total sample were performed. Triangulation was employed to develop a deeper understanding of the research topic.

### Results

Data from 26 patients (mean age 61.6 years [standard deviation 13.2]) were analysed. RAGT was highly accepted by patients and feasible, indicated by recruitment, retention, and adherence rates of 84.8% (95% confidence interval, CI 0.7–0.9), 92.2% (95% CI 0.7–1.0) and 94.0% (95% CI 91.4–96.2), respectively. Goal attainment was high, and only mild

**Data Availability Statement:** All data generated or analysed during this study are included within the manuscript itself and its supporting material.

**Funding:** Tyromotion GmbH, Graz, Austria A LEXO Gait Trainer was provided for the Clinic for Rehabilitation Münster to be able to conduct the study. No financial or other donations were given to any personnel involved in the study. The funders had no role in study design, data collection and analysis, decision to publish, or preparation of the manuscript.

**Competing interests:** The authors have declared that no competing interests exist.

adverse events occurred. Improvements in walking speed (10-Metre Walk Test, effect size r = 0.876), walking distance (6-Minute Walk Test, r = 0.877), functional mobility (Timed Up and Go, r = 0.875), gait distance (r = 0.829) and number of steps (r = 0.834) were observed. Four themes were identified: familiarising with RAGT; enjoyment and acceptance through a trusting therapeutic relationship; actively interacting; and minimising dissatisfaction.

## Discussion

Sufficiently powered randomised controlled trials are needed to validate our results.

## Trial registration

German Clinical Trials Register, DRKS00027887.

## Background

Walking disorders represent a relevant symptom in patients with neurological diseases. In multiple sclerosis (MS), symptoms such as weakness and spasticity, loss of proprioception, balance and coordination, cognitive and visual dysfunction, fatigue and pain can contribute to walking impairment [1]. In patients with Parkinson's disease (PD), mobility may be affected by rigidity, hypo- and bradykinesia, flexion posture and postural instability [2]. After stroke, hemiparetic spastic walking impairment is a common symptom in about 75% of patients [3]. Hereditary ataxias are primarily characterised by progressive motor and speech coordination deficits and unsteady walking with an increased step width [4]. In patients with spastic paraplegia or tetraplegia due to spinal cord injury (SCI), polyneuropathies, and motor neuron disease, walking dysfunction is also a major aspect of patients' disability leading to limitations in their daily lives [5]. Numerous studies have found that walking disability contributes to depression and anxiety and severely impairs social participation and health-related quality of life (HRQoL) [1, 6]. In addition, relevant studies have highlighted the relationship between walking disability and unemployment in patients with neurological diseases [7, 8].

Walking and balance disorders can be treated through intensive and repetitive task-oriented training [9, 10]. Applying the principles of massed (focused), high-dose, task- specific training with individualized progressive adjustment of difficulty considering patient's performance, implicit and explicit feedback, robot-assisted gait training (RAGT) is used to address walking impairment [11–13]. Systematic reviews have shown that RAGT benefits gait in patients with PD and MS [14–16] and improves walking ability, walking speed and distance in patients after stroke [17]. However, to the best of our knowledge, no study has investigated the feasibility and acceptability of RAGT in patients with common and rare neurological disorders using a mixed methods approach. Aims of this study were therefore to investigate the feasibility, acceptability, goal attainment and preliminary effects of RAGT in patients after stroke, with MS, PD, spastic paraplegia or tetraplegia, hereditary ataxia, inflammatory demyelinating polyneuropathies and motor neuron disease. Further aims were to explore the intervention context and process.

## Materials and methods

### Study design

This study was a prospective convergent mixed methods feasibility study [18] based on the paradigm of pragmatism to investigate the research question with a plurality of views and

methods [19]. Triangulation was used to gain a deeper understanding of the research topic [20]. The Good Reporting of a Mixed Methods Study (GRAMMS) framework [21] was used (S1 Table).

## Participants

**Sample size.** Based on the study design and primary outcome, no formal sample size calculation was carried out (see S1 File for the study protocol). Instead, the decision regarding the number of patients included in the study was influenced by the varying prevalence rates of the investigated diseases, resulting in a total sample size of 28.

**Inclusion criteria.** Using maximum variation sampling, 28 patients were enrolled from 1.2.2022 to 18.10.2022 at the Clinic for Rehabilitation Münster, Austria. Inclusion criteria were: adults aged 18–99 years; any sex; any ethnicity; diagnosis of ischaemic or haemorrhagic stroke confirmed using the German Society of Neurology (2017) Guideline [22]; MS according the revised McDonald criteria valid at the time of diagnosis [23, 24]; PD using the UK Brain Bank criteria [25]; spastic para- or tetraplegia caused by any type of spinal cord lesion (SCI) according to the German Society of Neurology (DGN 2021) Guideline [26] or current NICE Guideline (2016) [27]; mild to moderate hereditary (spinocerebellar) ataxia (SCA) using the Ataxia Medical Guidelines 2016 or earlier versions, developed by Ataxia UK, London [28]; acute or chronic inflammatory demyelinating polyneuropathy (PNP) e.g., due to Guillain-Barré syndrome or chronic inflammatory demyelinating polyneuropathy following the corresponding European Federation of Neurological Societies/Peripheral Nerve Society Guideline [29]; motor neuron disease (MND) e.g., amyotrophic lateral sclerosis (ALS) using the revised World Federation of Neurology El Escorial Criteria [30, 31]; at a clinically stable phase of the disease; ability to participate in this study; walking impairment (Functional Ambulation Categories (FAC) [32] score 0–4); intact cognitive function (Mini Mental State Examination (MMSE) [33] score of ≥24/30 points); German speaking verbally and written language.

**Exclusion criteria.** Patients were excluded if they met any of the following criteria: concomitant disease such as malignant disease, other severe neurological, orthopaedic or psychiatric disorders; acute, pronounced pain; joint contracture or arthrodesis; severe spasticity in or consolidated fractures of the lower extremity; body weight exceeding 180 kg; body height <100 cm or >200 cm; increased risk of bone fracture; osseous or joint instability; cardiac contraindications; high-grade ataxia or apraxia; skin lesions in areas which come in contact with the device or harness system; inadequately treated epilepsy; or pregnancy.

**Recruitment.** Patients who met the eligibility criteria were informed verbally and in writing about the study by the principal investigator, along with an invitation to participate in the study. Upon acquiring informed written consent, screening for cognitive impairment was carried out. Patients who passed the screening were subsequently enrolled in the study.

## Intervention

All patients received RAGT using a mechatronic end-effector robotic device with body weight support (LEXO® robotic gait trainer, Tyromotion, Graz, Austria). RAGT was performed as part of multidisciplinary inpatient rehabilitation for 45 minutes, 4x/week, for 4 weeks and supervised 1:1 by a trained physiotherapist. Patients could choose between a harness system or a saddle to ensure the required body-weight support (S1 Fig). Throughout the training session, augmented performance feedback was delivered via a computer screen positioned in front of the patient. This feedback was generated from the robot's gait parameter measurements. A standard RAGT protocol was used to record configurations, results, breaks and other important information (S2 File). The training was adapted according to the individual performance

level of the patient. The first step was to reduce the body-weight support as tolerated by the patient, then gradually increase the cadence and step length. During the 45 min training session, patients could rest at any time. Physical support at the ankle or knee joint was provided by the physiotherapist as needed, to assist muscle activation and enhance a physiological gait pattern. Adverse events and safety issues related to the training were monitored.

## Data collection

At baseline, age, sex, years of education, and type and date of diagnosis were extracted from patients' medical charts. Disease specific information were collected as follows: Impairment caused by stroke was assessed using the National Institutes of Health Stroke Scale (NIHSS) [34]; MS-related disability with the Expanded Disability Status Scale (EDSS) [35]; disability and impairment in PD utilising the Unified Parkinson's Disease Rating Scale (UPDRS) [36] and Hoehn & Yahr Scale (H&Y) [37]; classification of the acquired impairment in SCI according to the American Spinal Injury Association (ASIA) Impairment Scale (AIS) [38] and neurological level of injury; impairment caused by SCA by means of the Scale for the Assessment and Rating of Ataxia (SARA) [39, 40]; disability caused by Guillain-Barré syndrome utilising the Guillain-Barré syndrome disability scale (GBSDS) [41]; peripheral neuropathy using the Overall Neuropathy Limitations Scale (ONLS) [42]; and functional status in patients with ALS using the Amyotrophic Lateral Sclerosis Functional Rating Scale, revised version (ALS-FRS-R) [43].

**Primary outcome.** The feasibility of the methods and the potential for conducting a larger trial was explored using predetermined criteria. This involved monitoring monthly recruitment, retention, and adherence rates in relation to the RAGT intervention. Using a structured log, adherence to the RAGT was documented, capturing instances of non-adherence and reasons for discontinuation (e.g., lack of interest). Instances of participants not being retained in the study were also documented, with the reason e.g., withdrawal of consent. Any adverse events and side effects were systematically recorded and evaluated using a formal log including the date, type of adverse event, severity classified into serious/non-serious, causality differentiated into related/non-related with the interventions, and actions to restore or improve the patient's wellbeing and outcome.

Acceptability of the intervention was assessed using a 5-point Smiley Likert Scale asking 4 predefined questions on the acceptance of the intervention, with lower numbers representing higher acceptance/satisfaction: "How did you like the training?"(1); "How well did you feel during the training?"(2); "How satisfied were you with your performance?"(3); "How satisfied were you with your results?"(4).

The predetermined criteria for feasibility success were:

a. a target recruitment rate of 3 patients per month or 80%,

b. a target retention rate of 85%,

c. a target minimum adherence rate of 75% to the RAGT interventions (12 sessions out of a maximum of 16),

d. high safety of the intervention, no severe adverse events related with the study intervention,

e. high acceptability of the intervention.

**Secondary outcomes.** Secondary outcomes were walking speed as assessed by the 10-Metre Walk Test (10MWT) [44]; walking distance measured by the 6-Minute Walk Test

(6MWT) [45]; functional mobility using the Timed Up and Go (TUG) [46]; dynamic balance measured by the Four-Square Step Test (FSST) [47]; fall rate assessed by the number (percentage) of fallers/non-fallers; walking ability using the functional ambulation categories (FAC) [32]; and dynamic balance assessed by the Functional Gait Assessment (FGA) [48]. In non-ambulatory patients, trunk movement was assessed using the Trunk Control Test (TCT) [49] and the limits of stability regarding trunk movements utilising the Modified Functional Reach Test (MFRT) [50]. Additionally, concerns about falling were measured using the Falls Efficacy Scale-International (FES-I) [51]; fatigue employing the Fatigue Severity Scale (FSS) [52]; depression assessed using the Beck Depression Inventory, second edition (BDI-II) [53]; and HRQoL utilising the 5-level EQ-5D version (EQ-5D-5L) [54]. The amount of individual and predefined goal achievement was evaluated using Goal Attainment Scaling (GAS) [55]. Details on the administration and psychometric properties of standardised assessments are provided in S3 File. Gait parameters (gait speed, gait distance, number of steps and step length) during the RAGT intervention were recorded by the LEXO® robotic gait trainer. Outcome data were collected by trained neurologists, psychologists, and physiotherapists at baseline and post-intervention.

In addition, semi-structured observations of RAGT were performed by trained physiotherapist-researchers in weeks 1 and 4 to understand the intervention context and process.

## Observations of the RAGT intervention

Two observations of the RAGT intervention were conducted to understand the intervention context and process, recognise patterns, and move beyond selected outcome measures (S2 Table). This was considered of clinical relevance because high acceptability may help increase motivation and adherence in patients. Using a combined structured and unstructured observation protocol and taking notes, in weeks 1 and 4 RAGT sessions were observed by trained and experienced physiotherapist-researchers holding a Master's or PhD degree to investigate the RAGT process and acceptance [56]. The observer introduced herself to the study patients at the beginning of the RAGT session but stayed in the background as a silent observer for the remainder of the session.

The observation forms underwent a pilot phase involving 4 separate researchers across 5 sessions to evaluate congruence, practicability and to identify and correct any sources of error [57]. Following the pilot, an additional section for unstructured observations and two categories for setup and closure were incorporated. This expansion was intended to gather more comprehensive insights into the process.

**Reflexive statement.** To enhance trustworthiness and credibility, and acknowledge and reduce researcher bias, all researchers discussed their position, beliefs and prior experiences [58]. The researchers used a peer scrutiny and debriefing throughout the process of semi-structured and unstructured observations and qualitative data analysis to disclose personal biases. Qualitative data collection and analysis was described in detail to enrich dependability. Both observations and the participants' and therapists' verbatim expressions were included in the final report. A summary of the researchers' backgrounds involved in the qualitative strand of the study is presented in S3 Table.

## Ethics, governance, and study registration

This study was approved by the research ethics committee of the Medical University of Innsbruck, Austria (ref. 1403/2021; 21.01.2022). All patients signed the informed consent form prior to enrolment. Data collected in the context of this study were treated confidential and personal data pseudonymised. This clinical investigation followed the World Medical

Association Declaration of Helsinki (2013) [59], Guidelines of the International Organization for Standardization (ISO), ICH E6 Guideline for good clinical practice (2016) and Guideline for Clinical investigation of medical devices for human subjects—Good clinical practice (ISO/DIS 14155:2018). This study was prospectively registered with the German Clinical Trials Register on 26.01.2022 (DRKS-ID: DRKS00027887).

## Data analyses

**Statistical data analyses.** IBM SPSS Software, release 27.0 (IBM Corporation, Armonk, NY, USA) and GraphPad Prism 9, San Diego, California, were used for the data analysis. Statistical significance was defined as a two-tailed p-value 0.05. Descriptive statistics were used for the baseline demographic variables, primary and secondary outcome measures. Continuous data were checked for normal distribution using the Shapiro Wilk Test and visualised using Q-Q plots and histograms. Frequencies (percentage) were presented for count data. Medians (minimum—maximum) were reported for ordinal data or continuous data showing a non-normal distribution, and the mean (standard deviation [SD]) was used for continuous data. The number (percentage) of patients, who reached a clinically significant improvement in walking speed of at least 0.14 m/s on the 10MWT [60] was estimated, likewise, the number (percentage) of patients was calculated, who showed a clinically significant improvement in walking distance of at least 34.4 metres on the 6MWT [61]. Spearman's rank correlation coefficients were estimated between gait parameters and mobility tests (i.e., timed walking, mobility, and dynamic balance tests) and presented using a heatmap. Effect sizes were determined using the rank-biserial correlation coefficient (r) derived from the Wilcoxon signed-rank test. This coefficient offers insights into the strength and direction of association between paired samples, specifically the baseline and post-intervention measures in this study [62]:

$$r = Z/\sqrt{n} \tag{1}$$

where Z was the the Wilcoxon signed rank test Z-score and n was the total number of observations on which Z was based [61]. According to Cohen (1988), a correlation coefficient of 0.10 was considered a weak correlation; 0.30 a medium correlation, and 0.50 or higher a strong correlation [63].

With respect to the primary outcome of feasibility, the recruitment rate (%) was estimated by dividing the number of patients who consented by the N of eligible patients, multiplied by 100. The retention rate (%) was determined by dividing the N of patients who completed the study by the N of the total sample, times 100. The adherence rate (%) was calculated by dividing the N of RAGT sessions performed by the patients, divided by the N of the scheduled RAGT training sessions over the 4 week study period (16x), times 100 [64]. The recruitment, retention, and adherence rates were calculated with their 95% confidence interval (CI) according to the Wilson 'score' method cited by Newcombe [65]; when the proportion was close to 0 or 1, a Poisson approximation as described by Brown and colleagues was used [66].

**Qualitative data analysis.** Inductive reflexive thematic analysis (TA) according to Braun and Clarke [66] was used to analyse the observation data. Thematic analysis is distinguished by its inherent structure, and we adhered to the six analysis phases outlined by Braun & Clarke [67, 68]. Phase 1 involved familiarising ourselves with the dataset through thorough and critical reading. In phase 2, using f4analyse (Dr. Dresing & Pehl GmbH, Marburg, Germany) qualitative analysis software, two independent researchers worked systematically through the data set, identifying initial codes that held relevance or significance to the research question. These codes were discussed among the two coders and two additional researchers until a consensus was reached. Codes underwent multiple revisions and were cross-checked against the entire

dataset. Moving into the third phase, two researchers independently began drafting preliminary themes, designed to encapsulate clusters of codes sharing common meanings. Advancing to the fourth phase, these themes were then discussed among the four researchers, refined, and subjected to testing. The subsequent stage (phase 5) focused on enhancing, naming, and defining the themes. At the conclusion of this process, the themes were established and presented in written form alongside pertinent data excerpts, enriching overall understanding. The process was not strictly linear; instead, it was marked by iterative cycles involving multiple rounds of revisions, refinements, deletions, and the emergence of new codes and themes. Throughout this iterative journey, the codes and themes were consistently evaluated through peer scrutiny and debriefing, solidifying the trustworthiness and credibility of our findings [69], involving colleagues in the research process to offer feedback, insights, and critical evaluation. To ensure consensus, the evolved themes and descriptions underwent extensive discussion among four female researchers. The themes and descriptions were crafted from the researchers' perspective, informed by their backgrounds in neurology and physiotherapy.

## Results

### Participants' characteristics

Twenty-eight patients were enrolled, thereof 26 completed the study (10 women, 16 men). See Fig 1 for a flow chart. The patients' mean age was 61.6 (SD 13.2) years. At baseline, 23 patients were ambulatory and 5 were non-ambulatory, 5 patients were fallers (19.2%) and 21 were non-fallers (80.8%). Detailed demographic and disease related data are presented in Table 1.

### Primary outcome

The recruitment rate was 87.88% (95% CI 70.86–96.04; or 3–4 patients recruited per month), the retention rate 92.86% (95% CI 0.75–0.99) and the adherence rate 94.23% (95% CI 91.42–96.19). For RAGT, none of the patients chose the saddle for body-weight support, they felt more comfortable using the harness system. Only anticipated mild and transient adverse events were reported including muscle fatigue (n = 19), musculoskeletal pain (n = 5), sensory symptoms (n = 3), and a fall (n = 1).

Median (range) pleasure rating on the 5-point Smiley Likert Scale during the RAGT was 1 (1–2). Feeling well during RAGT was rated median 1 (1–3), satisfaction with one's performance 2 (1–3) and satisfaction with one's results 1 (1–4), with lower scores indicating higher pleasure, wellbeing, and satisfaction respectively. These numbers indicated that the predetermined feasibility criteria were met.

### Secondary outcomes

From baseline to post-intervention, there was an improvement in walking speed, walking distance, functional mobility, and dynamic balance in all patients (Fig 2 and S2 Fig and S4 Table). Sixteen out of 28 patients (57.1%) reached a clinically significant improvement in walking speed of ≥20% and 12 patients (42.9%) showed a clinically significant improvement in walking distance exceeding 20%. Compared to baseline, with 5 patients reporting a fall, at post-intervention only one patient remained a faller (3.8%) and 25 were non-fallers (96.2%). Overall, walking abilities according to the FAC improved by median (range) 1 (0–3) points. Patients diagnosed with MND, SCI, and SCA demonstrated the smallest improvements in their walking capabilities, with gains of only 0.5 points.

In non-ambulatory patients, improvements in the limits of stability regarding trunk movements (MFRT) were observed, with large effect sizes for all subcategories (reaching forward,

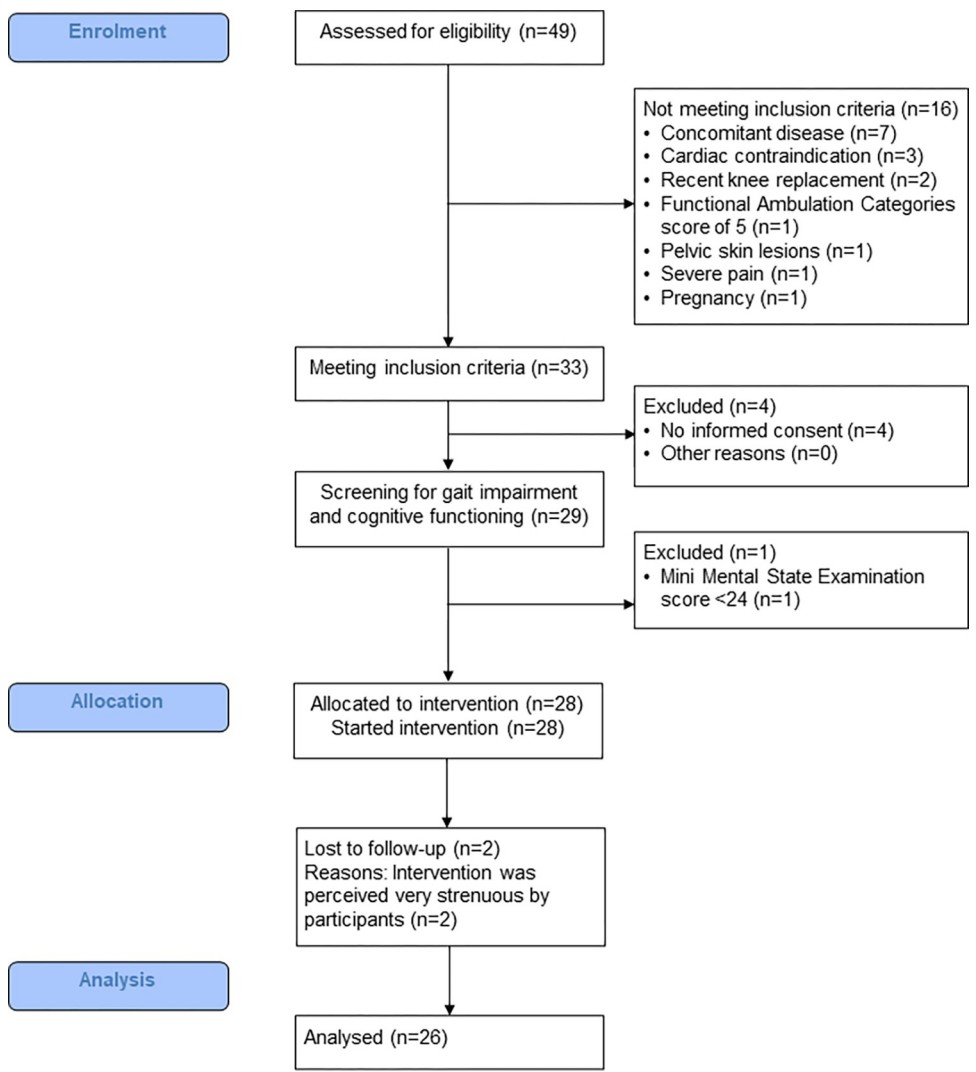

**Fig 1. Modified CONSORT flow diagram.**

r = 0.663; reaching right, r = 0.905 and reaching left, r = 0.921). There were however no changes in aspects of trunk movement as assessed by the TCT. Improvements in concerns about falling were seen in most patients, with a large effect size of r = 0.772. Furthermore, improvements in fatigue (r = 0.544), depression (r = 0.647) and HRQoL (r = 0.579, 0.404) were found, although some patients reported a worsening in fatigue, depression, or their overall health condition (Fig 3 and S4 Table). Patients were able to achieve individual and predefined goals "somewhat more than expected" as shown by a median (minimum—maximum) score difference on the GAS of 1 (-2 to 2).

A, ambulatory patients; BDI-II, Beck Depression Inventory second edition; BL, baseline; EQ-5D-5L Index, 5-level EQ-5D version Index; EQ-5D-5L VAS, 5-level EQ-5D version visual analogue scale; FES-I, Falls Efficacy Scale-International; FSS, Fatigue Severity Scale; MND, motor neuron disease; MS, multiple sclerosis; NA, non-ambulatory patients; PD, Parkinson's disease; PI, post-intervention; PNP, acute or chronic inflammatory demyelinating polyneuropathy; SCA, spinocerebellar ataxia; SCI, spinal cord injury (spastic para- or tetraplegia).

**Table 1. Demographic and disease specific data.**

| Parameter | Stroke, subacute, NA (n = 5) | Stroke, chronic, A (n = 5) | MS (N = 5) | PD (n = 3) | MND (n = 2) | SCI (n = 2) | SCA (n = 2) | PNP (n = 2) | Total sample (n = 26) |
|---|---|---|---|---|---|---|---|---|---|
| Age (years)[1] | 68.6 (1.3) | 61.1 (2.2) | 51.6 (8.0) | 65.2 (6.0) | 60.0 (30.1) | 55.6 (9.9) | 55.7 (17.7) | 60.5 (17.3) | 61.6 (13.2) |
| Sex M: F[2] | 4: 1 | 3: 2 | 0: 5 | 2: 1 | 2: 0 | 1: 1 | 2: 0 | 2: 0 | 16: 10 |
| Education (years)[3] | 12.0 (11.0–18.0) | 12.0 (8.0–12.0) | 14.0 (9.0–25.5) | 14.0 (12.0–17.0) | 14.0 (11.0–17.0) | 18.0 (12.0–24.0) | 12.5 (12.0–13.0) | 14.2 (12.5–16.0) | 12.0 (8.0–25.5) |
| Disease duration (years)[3] | 0.2 (0.1–0.3) | 2.2 (0.4–5.7) | 21.2 (5.0–34.7) | 0.9 (0.3–5.4) | 15.0 (0.3–29.7) | 9.9 (1.5–18.3) | 8.4 (7.6–9.3) | 3.5 (0.4–6.6) | 4.8 (0.1–34.7) |
| MMSE[3]* | 30.0 (26.0–30.0) | 30.0 (26.0–30.0) | 27.0 (24.0–29.0) | 29.0 (27.0–30.0) | 27.0 (24.0–30.0) | 30.0 (30.0–30.0) | 29.0 (28.0–30.0) | 30.0 (30.0–30.0) | 29.0 (24.0–30.0) |
| FAC[3] | 0.0 (0.0–2.0) | 4.0 (3.0–4.0) | 4.0 (2.0–4.0) | 4.0 (4.0–4.0) | 4.0 (3.0–4.0) | 4.0 (4.0–4.0) | 4.0 (4.0–4.0) | 4.0 (4.0–4.0) | 4.0 (0.0–4.0) |
| Impairment / disability[3] | NIHSS** 11.0 (9.0–17.0) | NIHSS** 10.0 (8.0–11.0) | EDSS** 6.0 (4.0–6.5) | UPDRS** 27.0 (22.0–30.0); H & Y** 1.0 (1.0–3.0) | ALS-FRS-R* 44.0 (43.0–44.0) | AIS** D (both); NLI L1 (both) | SARA** 10.0 (3.0–17.0) | GBSDS** 2.0 (2.0–2.0); ONLS** 5.0 (2.0–7.0) | |

A, ambulatory patients; AIS, ASIA (American Spinal Injury Association) Impairment Scale (grade); ALS-FRS-R, Amyotrophic Lateral Sclerosis Functional Rating Scale, revised version; EDSS, Expanded Disability Status Scale; F, female; GBSDS, Guillain Barré Syndrome Disability Score; H & Y, Hoehn & Yahr scale; M, male; MMSE, Mini Mental State Examination; MND, motor neuron disease; MS, multiple sclerosis; N, number; NA, non-ambulatory patients; NIHSS, National Institutes of Health Stroke Scale; NLI, neurological level of injury; ONLS, Overall Neuropathy Limitations Scale; PD, Parkinson's disease; PNP, polyneuropathy; SARA, Scale for the Assessment and Rating of Ataxia; SCA, spinocerebellar ataxia; SCI, spinal cord injury (spastic para- or tetraplegia); UPDRS, Unified Parkinson's Disease Rating Scale.

[1] Data represent mean (standard deviation).

[2] Frequency.

[3] Data represent median (minimum—maximum).

*The higher the score, the better the function.

**The higher the score, the worse the function.

Overall, the gait parameters of gait distance, number of steps, average step length, RAGT time, and maximum gait speed improved in all disease groups (S5 Table). For the relationship between the RAGT gait parameters and the timed walking tests, mobility, and dynamic balance tests, weak to moderate correlations were found (Fig 4). Fig 5 presents changes of the gait distance, number of steps, average step length, RAGT time and maximum speed from baseline to post-intervention in the total sample.

## Observations of the RAGT intervention

Throughout the intervention period, 51 observations of RAGT were carried out on 26 patients. Among them, 23 patients underwent two observations each. The field notes taken were converted into digital Word files for subsequent data analysis. From the material 56 codes were developed, which brought the key points into focus [70] (S6 Table). Finally, four themes were identified from the codes, as shown in Table 2, along with their corresponding quotations. The themes were not prioritised; their order reflects the sequence in which they were found.

## Integration of findings

Integration of quantitative and qualitative data was performed by two researchers (IH and BS), showing that RAGT is feasible and highly accepted among patients with neurological disorders. Fig 6 visualises the individual positive and negative aspects of the RAGT and how they interacted. Due to the high adaptability of the device, the training parameters could be

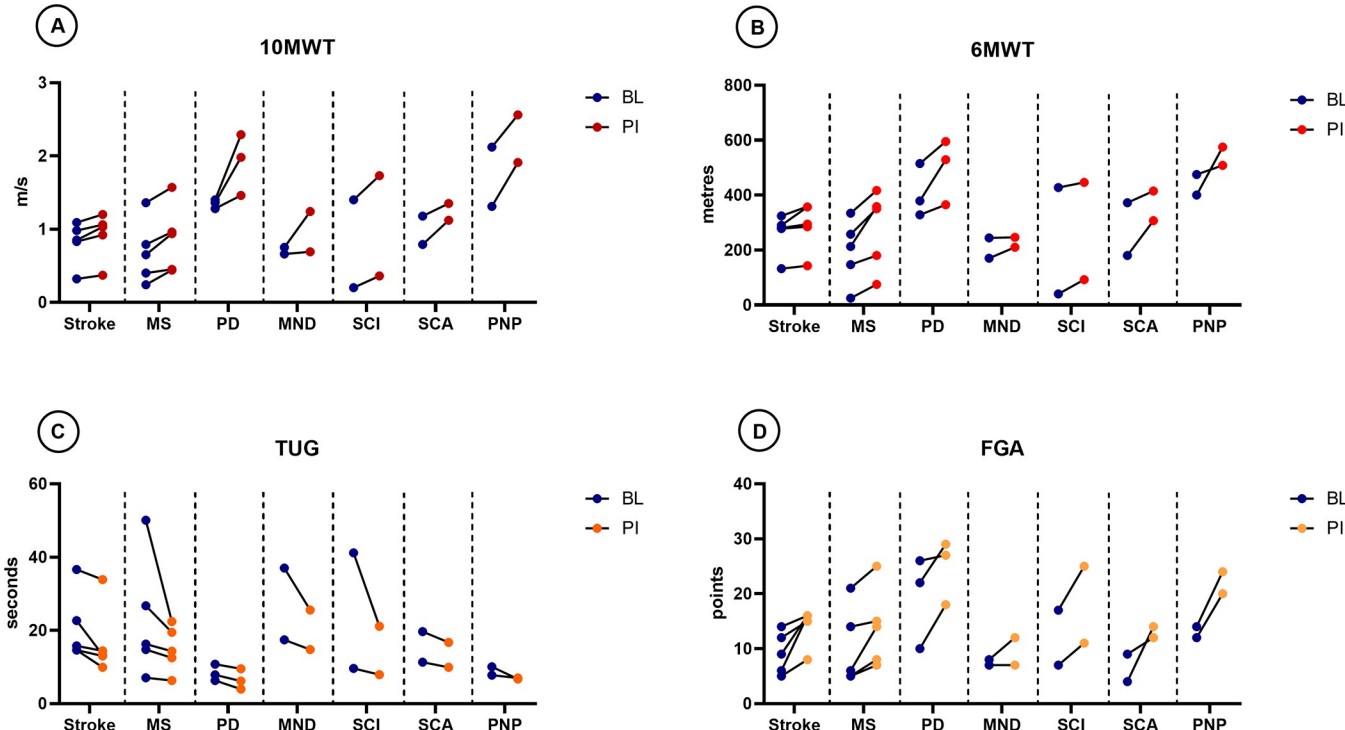

**Fig 2. Secondary outcomes.** The before-after graphs represent individual patients' walking performance at baseline (blue dots) and post-intervention (red or orange dots) on the (**A**) 10MWT, (**B**) 6MWT, (**C**) TUG, and (**D**) FGA. With the 10MWT, 6MWT and FGA, an increase in scores represents improvement. With the TUG, decrease in duration indicates improvement. 10MWT, 10-Metre Walk Test; 6MWT, 6-Minute Walk Test; BL, baseline; FGA, Functional Gait Assessment; MND, motor neuron disease; MS, multiple sclerosis; PD, Parkinson's disease; PI, post-intervention; PNP, acute or chronic inflammatory demyelinating polyneuropathy; SCA, spinocerebellar ataxia; SCI, spinal cord injury (spastic para- or tetraplegia); TUG, Timed Up and Go.

adjusted to fit the patients' needs. Moreover, quantitative data showed that walking ability, walking speed, mobility, dynamic balance, gait distance, RAGT time, average step length and maximum gait speed improved over the 4-week intervention period. Furthermore, improvements in concerns about falling, fatigue, mood, and HRQoL were reported. The therapeutic relationship between patients and therapists was observed to be a core component of a satisfying and self-determined training for the patients, where they were able to unleash their full potential.

## Discussion

This study investigated the feasibility, acceptance, and preliminary effects of RAGT among people with different neurological diseases. Results showed that a 4-week intervention embedded into routine care was feasible and highly accepted by the patients. In our study, high recruitment, adherence, and retention rates of 87.9%, 94.2% and 92.3% respectively, were observed. Given the lack of evidence on recruitment, adherence, and retention rates in RAGT studies, we compared our results with other rehabilitation studies. With a home-based physical activity programme in older adults at risk of Alzheimer's disease, adherence and retention rates were 91.7% and 97.2% respectively [71]. An 80% retention rate was seen with a 6-week rhythmic-auditory cued gait and balance programme in people after stroke [72]. A review that investigated recruitment and retention of participants in 151 randomised controlled trials (RCT) funded and published by the United Kingdom Health Technology Assessment Programme showed a median retention rate of 89.0% [73], comparable the present study. We

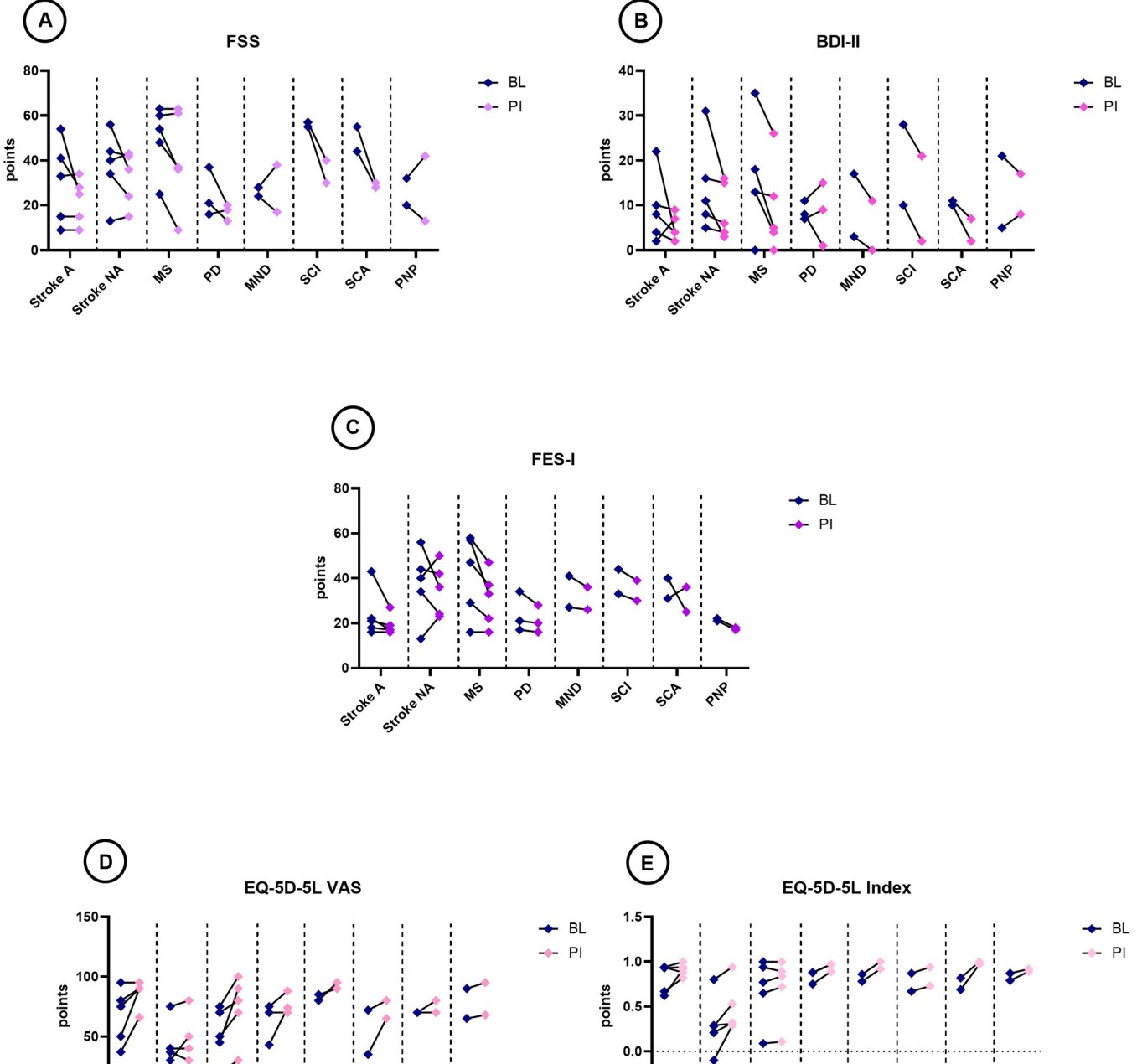

**Fig 3. Results from secondary outcomes of questionnaires.** The before-after graphs represent individual patient's scorings at baseline (blue squares) and post-intervention (lilac or pink squares) on the (**A**) FFS, (**B**) BDI-II, (**C**) FES-I, (**D**) EQ-5D-5L VAS and (**E**) EQ-5D-5L Index. With the EQ-5D-5L VAS and EQ-5D-5L Index, an increase in scores represents improvement. With the FSS, BDI-II and FES-I, decrease indicates improvement.

suggest that several factors could have influenced adherence and retention in our and other studies. First, individual characteristics, such as motivation, perceived benefits, and expectations, supervision, integration into daily life, communication and feedback, self-efficacy and competence and patient's active role and goal setting may have impacted adherence and

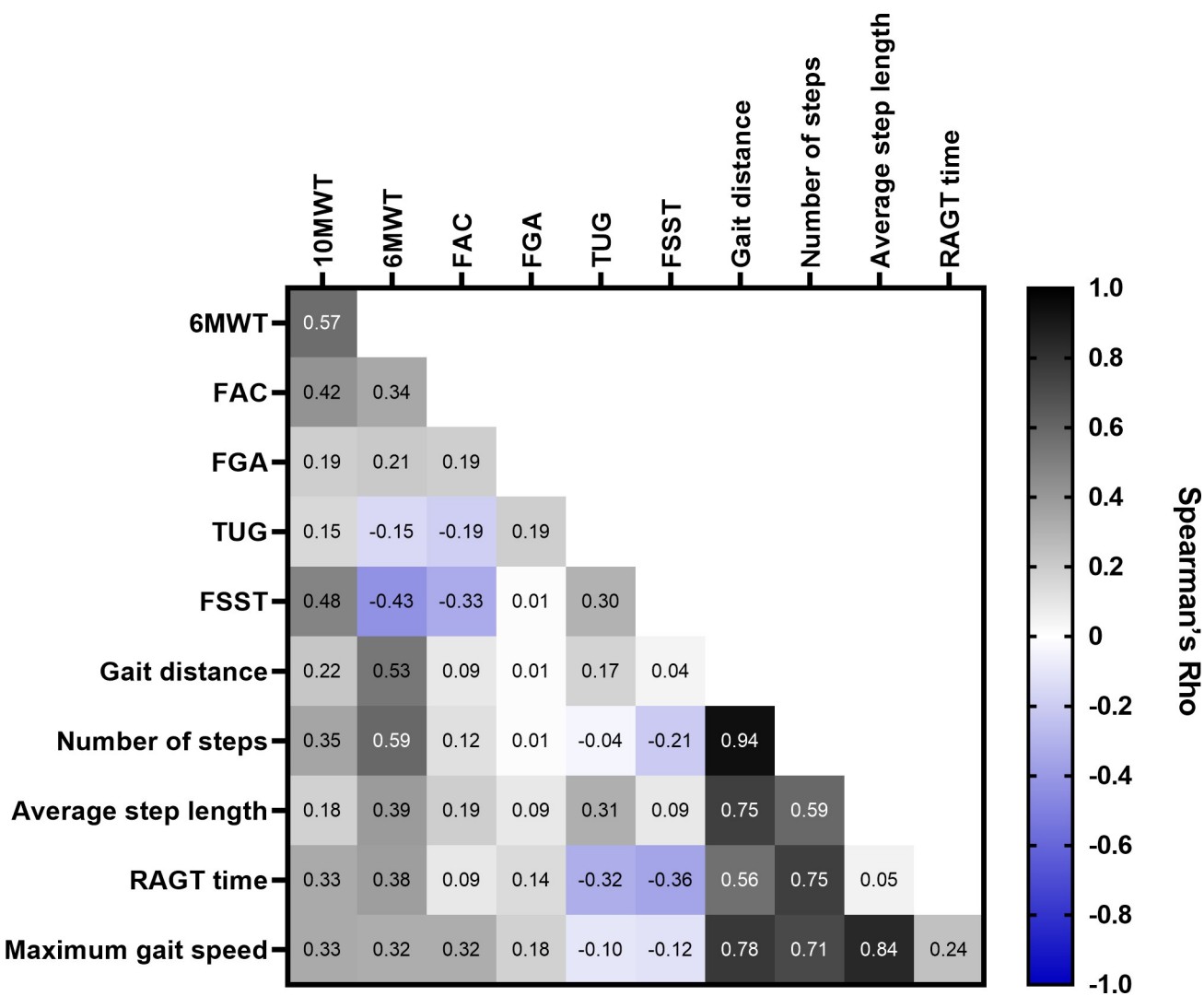

**Fig 4. Correlations of gait parameters with walking, mobility, and dynamic balance.** The heatmap presents Spearman's rank correlation coefficients (ranging from -1 to +1) of gait parameter measures and walking assessments, where greater numbers (Spearman's rho correlation coefficients), and darker blue and black fields signify stronger positive or negative correlations. 10MWT, 10-Metre Walk Test; 6MWT, 6-Minute Walk Test; FAC, Functional Ambulation Categories; FGA, Functional Gait Assessment; FSST, Four Square Step Test; RAGT, robot-assisted gait training; TUG, Timed Up and Go.

retention [74], particularly in those with more severe walking impairment. Patients with higher motivation levels, a positive attitude towards training, and a clear understanding of the potential benefits of RAGT are likely more inclined to actively participate in and adhere to a RAGT programme, ensuring continued engagement [74]. Second, the design of the RAGT intervention itself could have influenced adherence and retention. Patient engagement and adherence may have been influenced by various factors, including the duration and frequency of sessions, customisation tailored to individual needs, the level of challenge and gradual progression in the training programme, as well as the availability of progress information and monitoring [74, 75]. Overall, in our study, only mild, short-lived adverse events were observed. These adverse events were of an expected nature and encompassed muscle fatigue, musculoskeletal pain, and sensory symptoms. Additionally, typical signs of exertion, such as heavy breathing, sweating, and skin reddening, were observed. Another end-effector study

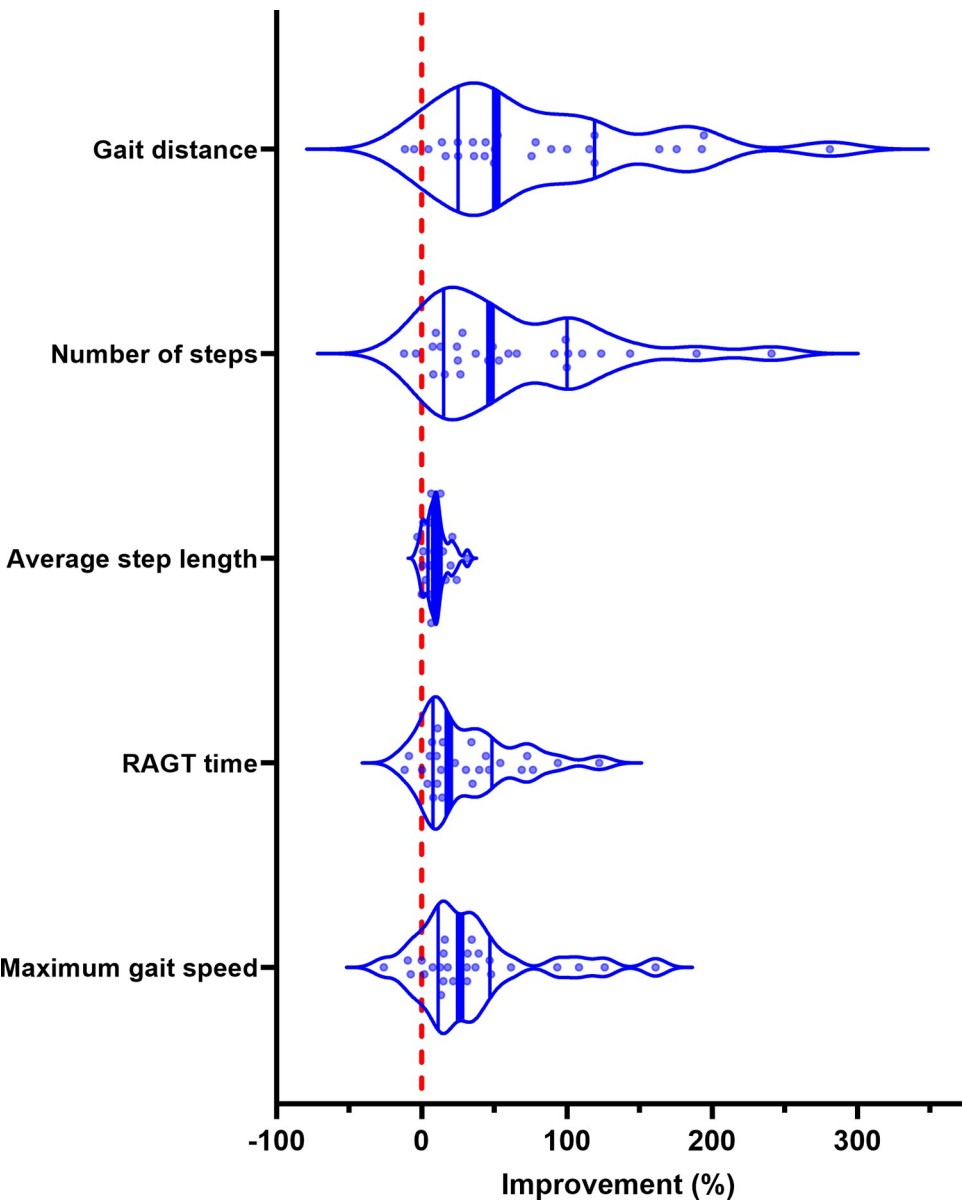

**Fig 5. Changes in gait parameters.** Violin plots present the changes in gait parameters of the study cohort. Medians are represented by bold vertical lines in the centre of the violins, 25. and 75. percentiles by thin vertical lines and ranges by the violin borders. Improvement is signified by a positive change (percentage) from baseline and worsening by a negative change (percentage), with a dashed red line indicating zero change. RAGT, robot-assisted gait training.

investigated in the feasibility of RAGT in patients with various neurological disorders and reasons for attrition [76]. Except saddle discomfort, main reasons for discontinuation comprised poor cooperation, pain, limitations in motion or need for excessive body-weight support [76]. This could be explained by the use of a saddle with the end-effector device in the cited study [76]. In contrast, participants in our study had the option to choose between a harness or a saddle, and they unanimously preferred the use of a harness. In general, technology-based interventions seem to be associated with higher adherence rates as compared to conventional therapies [77]. Third, adequate support and guidance from the therapists in our study could have played a crucial role in promoting adherence and retention. Provision of clear

**Table 2. Results of the thematic analysis.**

| Theme | Description of the theme | Quotations |
|---|---|---|
| **1**<br>**Familiarising with RAGT** | In the initial therapy sessions, RAGT may seem unfamiliar to patients. They often approach the training with a mixture of curiosity and a degree of nervousness. However, these feelings can be effectively addressed through the device's high adaptability and a proactive, flexible approach to organising the sessions. | "The first time it took a lot of getting used to, in terms of movement and intensity" (ID14, MS) |
| **2**<br>**Enjoyment and acceptance through a trusting therapeutic relationship** | Establishing a trustworthy therapeutic rapport between the patient and therapist, grounded in appreciative and transparent communication, providing comprehensive information about the device and training, including procedural instructions and therapeutic tasks, along with offering tactile assistance, all contribute to enhancing the acceptance of RAGT and the enjoyment and satisfaction derived from the training sessions. | "How cool is that—I'm the best! I like it, now I like myself again." (ID 13, stroke)<br>"The therapist explains every single step, points out everything to the patient in advance, is very approachable. She often asks how things are going, is encouraging and empathetic." (ID22, ataxia) |
| **3**<br>**Actively interacting** | Patients' active engagement throughout the training, including their involvement in preparation and closure, setting both short- and long-term goals, experiencing appropriately calibrated challenges, and exercising self-determination during therapy, collectively contribute to heightened patient satisfaction and a sense of personal responsibility. Consequently, this comprehensive approach results in the perception of therapy as more effective and beneficial. | "So especially with this device, with the LEXO, I already have gained more strength. And I take nice brisk steps with my right foot using the rollator." (ID26, stroke)<br>"I have more strength in my legs and the performance is better. I am happy I am at the LEXO because the other therapies feel less effective" (ID21, Parkinson) |
| **4**<br>**Minimising dissatisfaction** | Factors such as dissatisfaction with personal performance, insufficient effort, doubts regarding the efficacy of RAGT, low affinity between therapist and patient, and existing condition-specific symptoms that are intensified during RAGT, often accompanied by symptoms of exertion and muscle fatigue, have the potential to diminish the success of the therapy. As a result, it is important for therapists to proactively address these issues and optimise the outcome of the therapy, for example through strategies like incorporating breaks or adjusting the level of body weight support. | "I'm dissatisfied, last week I felt better" (ID3, MS)<br>"The patient states that it is more strenuous than yesterday. Puffs very hard after only a few minutes." (ID3, MS) |

instructions, ongoing feedback, encouragement, and addressing of concerns or barriers could have helped patients feel supported and motivated to continue with the RAGT [78, 79]. This was reflected by the findings from our qualitative analysis described below.

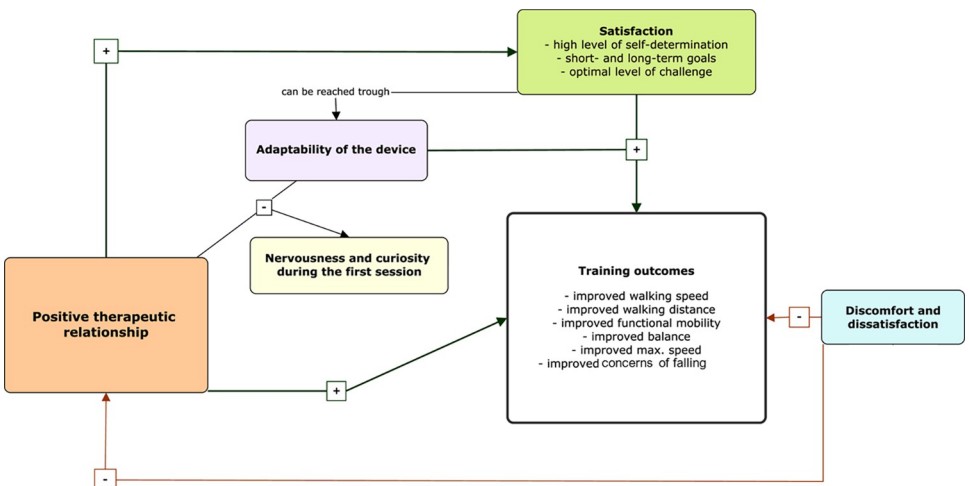

**Fig 6. Integration of qualitative and quantitative findings.** The concept-map illustrates the individual aspects, with plus signs denoting positive influences and minus signs negative influences on satisfactory RAGT.

Our quantitative results showed improvements in all walking, balance, and functional mobility related outcomes, as well as in concerns about falling, fatigue, depression and HRQoL across groups. We found large effect sizes related to changes in walking speed and distance, functional mobility, dynamic balance, gait distance, number of steps, RAGT time and concerns about falling in patients. Significant albeit weak to moderate correlations between RAGT parameters and mobility related outcome measures were seen. This suggests that RAGT is useful even within underexplored neurological populations, including patients with motor neuron disease and SCA.

In an RCT that compared RAGT with different amounts of weight support to conventional gait training in patients after stroke, improvements in walking speed and mobility were found [80]. Results aligned with those from a single-blind pilot RCT [81] on the effects of RAGT with virtual reality on motor and cognitive function in people with MS. A significant improvement in walking distance and HRQoL was observed, again, similar to the present study. We speculate that the end-effector-based gait training type used in our study helped to address pelvic obliquity in patients, by providing targeted assistance and guidance to promote proper alignment and symmetry [82]. The robotic device may have contributed to adjusting movement trajectories, enhancing the level of voluntary muscle engagement during the stance phase and applying forces to the body to encourage correct pelvic alignment [83, 84]. These suggestions align with findings from RCTs that investigated the differences between end-effector RAGT and treadmill training or conventional gait training in people with PD [85–88]. Results showed greater improvements in pelvic obliquity and the hip abduction-adduction in the RAGT group than in the treadmill group [86] and additionally, RAGT was superior to conventional gait training [85, 86]. However, no statistical superiority of RAGT over treadmill training in balance or walking speed was found [87]. This might be attributed to a lack of statistical power, resulting from 20–25 patients in each group.

Studies investigating the effects of RAGT in patients with SCA are still lacking. Our preliminary results indicated that patients with SCA may benefit from RAGT. These results are in line with two single case studies showing improvement in balance and walking in a 56-year-old male patient with hereditary ataxia of unknown type [89] and a 23-year-old woman with SCA [90]. Findings suggest that individuals with ataxia may benefit from the assistance, guidance, and feedback provided by the RAGT technology. As RAGT devices provide a more stable and controlled environment, the risk of falls is reduced. Consequently, patients can focus on their gait patterns responding to the real-time feedback on gait parameters from the robotic device, enabling them to make necessary adjustments and enhance their motor control and coordination during walking [91, 92]. This may refer particularly to patients with intact cognition as included in our study, who are able to understand and generate a physical response to visual feedback [92]. Agreeing with this proposition, an RCT including 19 patients after stroke or meningoencephalitis with ataxia demonstrated significantly greater improvements in walking ability and walking speed after end-effector type RAGT as compared to conventional physiotherapy [93]. Similarly, another small RCT has found statistically significant improvements in balance, functional independence in activities of daily living and general ataxia symptoms, but no differences between the RAGT and therapist-assisted gait training groups in patients with post-stroke ataxia [94].

Our descriptive results indicated that RAGT leads to improvements in walking, balance, mobility, and other outcomes in people with spastic paraplegia due to SCI. Relatedly, a meta-analysis showed that RAGT was superior to body-weight-supported overground or treadmill walking in improving walking speed in people with SCI [95]. In line with this, another prospective multi-centre study has shown significant improvements in walking speed following an end-effector-based gait training in people with SCI [96].

In our study, we observed an improvement in walking, balance, mobility, and further outcomes in people with MND following 4-week RAGT. As MND includes a group of progressive neurological disorders leading to spasticity, cramps, muscle weakness and wasting, we carefully adjusted the RAGT to the patients' functional status. This was based on meagre evidence showing that endurance or exercise training of moderate intensity can improve motor function in people with ALS [97, 98]. One study investigated the effects of ≥4-week RAGT on walking speed and distance and activities of daily living using a wearable exoskeleton in five patients with ALS [99]. Improvements in walking were found. Another small pilot study (n = 9; 3 drop-outs) explored the feasibility, tolerability, safety, and preliminary effects of an 8-week repetitive rhythmic exercise and body-weight supported treadmill gait training in patients with ALS [100]. Improvements in walking and fatigue were seen and the intervention was considered safe and tolerable while the feasibility of a larger study was not clear [100]. Except for fatigue, our results confirmed those of the two studies. The small sample sizes do however need to be considered.

Our qualitative analysis indicated that strong therapeutic connection played a central role in fostering high levels of acceptance and satisfaction with the RAGT (theme 2). This relationship should be built on mutual appreciation, open communication, provision of clear information about the device and the training, including instructions and therapeutic tasks, and offering tactile support. Our findings echoed a study from Miciak who identified a set of four key conditions for establishing a therapeutic relationship: being present, receptive, genuine, and committed as a therapist [101]. Furthermore, our results were in line with a retrospective observational study nested within an RCT showing that treatment outcomes of patients with lower back pain were consistently influenced by the quality of the therapeutic alliance [102]. A systematic review including 13 prospective RCTs, controlled clinical trials, and cohort studies has reported that a trustful therapeutic alliance was associated with increased treatment adherence and satisfaction and improved depressive symptoms and physical function [103]. These results imply that except involving highly repetitive task-oriented training, RAGT therapists should also be encouraged to strengthen the therapeutic alliance. Another longitudinal study investigated the turning points in the therapeutic relationship development in occupational and physical therapy [104]. From the perspective of each partner in a dyadic relationship, turning points are the pivotal moments in a relationship that have the most significant impact on its trajectory [105]. Findings from their study showed that progressing towards a goal, positive feedback, and a strong interpersonal affective bonding with the therapist can be relevant turning points generating positive emotions and favourable perceptions of relationship quality [104].

Further themes identified through thematic analysis were familiarising with RAGT (1), actively interacting (3) and minimising dissatisfaction (4). Familiarising with RAGT included the first session(s) and the accompanying feelings of unfamiliarity, curiosity, and a certain nervousness with RAGT. This process, as well as active interaction have previously been identified as key points in a qualitative analysis of patients' and therapists' experiences and perceptions of exoskeleton-based physiotherapy in stroke rehabilitation [106]. The intimidating reception of a device of this size was also described in a mixed-methods study in 10 individuals after stroke [107]. Nevertheless, patients' perception of receiving an effective and useful therapy overweighed any initial insecurities [106, 107]. Notably, several studies examining RAGT reported the presence of discomfort or dissatisfaction [76, 106, 107] as it was also observed in our study. This needs to be considered before allocating patients to RAGT, for this might lead to intensifying pre-existing symptoms (theme 4) or even discontinuation [76]. Our results indicate that therapists should therefore address patients' insecurities about effectiveness and dissatisfaction as best as possible by adjusting hardware and software parameters like body-weight support, but also give clear information on a basis of a trusted therapeutic relationship.

## Strengths and limitations

Through a mixed methods design, we were able to gather information on both the feasibility and preliminary effectiveness of RAGT and insight into the patients' experience in real-life RAGT. Nevertheless, there were limitations. Primarily, it is worth noting that during the 4-week intervention period, the patients did not receive RAGT as the sole intervention. This approach aligns with other studies conducted in an inpatient rehabilitation setting. Hence, it was not possible to attribute the effects solely to RAGT as the patients underwent a complex multimodal rehabilitation programme. Additionally, there was no follow-up testing to assess the sustainability of the results.

Another limitation was the absence of patient interviews at the conclusion of the intervention period, results of which may have been valuable and insightful. Nonetheless, the observations facilitated a deeper understanding of the patients' and therapists' behaviour and provided an opportunity to capture details during RAGT that the people themselves might have overlooked. This allowed the research team to draw conclusions on critical factors to be considered in RAGT and derive recommendations for tailoring RAGT to meet the individual needs of the patients.

## Conclusions

To our knowledge, there is a gap in the literature regarding the exploration of quantitative RAGT effects alongside patient acceptance and satisfaction using qualitative methods across various neurological conditions. Our study aims to fill this gap by offering valuable insights into understanding patients' needs and identifying necessary adjustments to ensure that RAGT is both enjoyable and maximally effective for patients.

The outcomes of our study imply the feasibility of utilising the LEXO® gait trainer for RAGT among patients with various neurological conditions. This is evidenced by robust recruitment, retention, and adherence rates. Furthermore, the findings underscore the safety of this approach, as indicated by the occurrence of anticipated mild and transient adverse events, alongside its initial effectiveness in enhancing RAGT outcomes among patients with diverse neurological conditions. Moreover, the majority of patients experienced the training positively and expressed satisfaction with their performance and outcomes. While our findings are promising, it is essential to conduct subsequent adequately powered RCTs to validate our results regarding the safety and efficacy of end-effector-based RAGT in patients with MND and SCA. Additionally, further mixed-methods studies are warranted to provide more nuanced insights into patients' perceptions of RAGT with end-effector devices and the influence of the therapeutic relationship on RAGT outcomes.

## Supporting information

**S1 Checklist.**
(DOCX)

**S1 Fig. LEXO training.** The figure shows the LEXO® gait trainer with harness system (**A**) and with the saddle (**B** and **C**). Reprinted from device training material under a CC BY license, with permission from Tyromotion GmbH, Graz, Austria, original copyright 2023. Written consent for publication was obtained from the person depicted in the pictures.
(TIF)

**S2 Fig. Results FSST and FAC.** The before-after graphs represent individual patients' walking and balance performance at baseline (blue dots) and post-intervention (yellow/green dots) on the (**A**) FSST and (**B**) FAC. A larger dot size indicates a higher number of patients achieving

the same score. With the FAC, an increase in scores represents improvement. With the FSST, a decrease in duration indicates improvement. A, ambulatory patients; BL, baseline; FAC, Functional Ambulation Categories; FSST, Four Square Step Test; MND, motor neuron disease; MS, multiple sclerosis; NA, non-ambulatory patients; PD, Parkinson's disease; PI, post-intervention; PNP, acute or chronic inflammatory demyelinating polyneuropathy; SCA, spinocerebellar ataxia; SCI, spinal cord injury (spastic para- or tetraplegia).
(TIF)

**S1 File. Study protocol.**
(PDF)

**S2 File. LEXO training protocol.**
(DOCX)

**S3 File. Description and psychometric properties of assessments.**
(DOCX)

**S1 Table. GRAMMS checklist.**
(DOCX)

**S2 Table. Observation protocol.**
(DOCX)

**S3 Table. Reflexivity of the researchers.**
(DOCX)

**S4 Table. Results of secondary outcomes.**
(DOCX)

**S5 Table. Results of gait parameters.**
(DOCX)

**S6 Table. Coding tree.**
(DOCX)

## Acknowledgments

We extend our heartfelt gratitude to all the patients who participated in this study. Additionally, we express our appreciation to the therapists at the Clinic for Rehabilitation Münster, Bianca Slamik, Markus Rendl, Stefan Klinger, and Katharina Lukasser, as well as the head and deputy head of the therapy department, Andreas Mühlbacher and Barbara Linert. We would also like to acknowledge the exceptional efforts of the therapy scheduling team around Sylvia Radinger.

## Author Contributions

**Conceptualization:** Michaela Stampfer-Kountchev, Christian Brenneis, Barbara Seebacher.

**Data curation:** Sarah Mildner, Barbara Seebacher.

**Formal analysis:** Sarah Mildner, Bianca Slamik, Barbara Seebacher.

**Funding acquisition:** Christian Brenneis, Barbara Seebacher.

**Investigation:** Isabella Hotz, Michaela Stampfer-Kountchev, Bianca Slamik, Christoph Blättner, Elisabeth Türtscher, Franziska Kübler, Clemens Höfer, Johanna Panzl, Michael Rücker.

**Methodology:** Christian Brenneis, Barbara Seebacher.

**Project administration:** Isabella Hotz, Barbara Seebacher.

**Supervision:** Michaela Stampfer-Kountchev, Christian Brenneis, Barbara Seebacher.

**Writing – original draft:** Isabella Hotz, Barbara Seebacher.

**Writing – review & editing:** Sarah Mildner, Michaela Stampfer-Kountchev, Bianca Slamik, Christoph Blättner, Elisabeth Türtscher, Franziska Kübler, Clemens Höfer, Johanna Panzl, Michael Rücker, Christian Brenneis, Barbara Seebacher.

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
