## [Editor Report · Decision Letter 0]

20 Oct 2023

PONE-D-23-31858

Robot-assisted gait training in patients with various neurological diseases: a mixed methods feasibility study

PLOS ONE

Dear Dr. Seebacher,

Thank you for submitting your manuscript to PLOS ONE. After careful consideration, we have decided that your manuscript does not meet our criteria for publication and must therefore be rejected.

I am sorry that we cannot be more positive on this occasion, but hope that you appreciate the reasons for this decision.

Kind regards,

Dimitrios Sokratis Komaris, Ph.D

Academic Editor

PLOS ONE

**Additional Editor Comments:**

Specifically:

I checked the provided protocol, and “5 patients with severe stroke in the subacute phase will be included in this study, … and 2 patients with motoneuron disease” should have been included in this study. Please clearly mention in the manuscript if this was indeed the case in the conducted study (ie, to avoide selective reporting).

Additionally, the study sample size is not properly justified, nor is the study a controlled trial as per the CONSORT requirements with appropriate randomisation procedures. Therefore, I cannot consider this submission as a clinical trial.

- - - - -

---

## [Author Response · Author response to Decision Letter 0]

8 Jan 2024

PONE-D-23-31858

Robot-assisted gait training in patients with various neurological diseases: a mixed methods feasibility study 

Dear Editor,

We extend our gratitude for your reconsideration of our manuscript. The dedication to the editorial process thus far is genuinely appreciated.

In response to the concerns raised by the Academic Editor, as outlined in your recent communication, we have prepared a comprehensive rebuttal letter, providing a point-by-point response. Unfortunately, as our manuscript has not undergone external peer review, we are currently unable to furnish specific details regarding revisions made since the initial submission. There we cannot provide a manuscript with tracked changes. Since it is a requirement of your editorial manager system to upload a manuscript with tracked changes, we have uploaded the original manuscript twice.

Point-by-point response to the Academic Editor's comments

Point 1: I checked the provided protocol, and “5 patients with severe stroke in the subacute phase will be included in this study, … and 2 patients with motoneuron disease” should have been included in this study. Please clearly mention in the manuscript if this was indeed the case in the conducted study (ie, to avoid selective reporting).

Response: We can confirm that our study included these groups, as indicated in Table 1. Patients in the subacute phase after a stroke had a median (minimum - maximum) Functional Ambulation Categories score of 0 (0-2), with a score of 1 representing a nonfunctional ambulator, and scores of 1-2 indicating an ambulator who is dependent on physical assistance. Patients with motor neuron disease exhibited mild walking impairment.

Points 2 and 3: Additionally, the study sample size is not properly justified (Point 2), nor is the study a controlled trial as per the CONSORT requirements with appropriate randomisation procedures. Therefore, I cannot consider this submission as a clinical trial. (Point 3)

Response to Point 2: Our study was, in fact, a mixed methods feasibility study. In the materials and methods section, we provided the following rationale for our sample size:

“Based on the study design and primary outcome, no formal sample size calculation was carried out. Instead, the decision regarding the number of patients included in the study was influenced by the varying prevalence rates of the investigated diseases, resulting in a total sample size of 28. Considering the study design, methods, and objectives, no additional patients were included to account for an anticipated attrition rate of 10%.”

We reviewed other mixed methods feasibility studies published in PLOS ONE, and none of these studies provided a formal sample size calculation or mentioned sample size considerations. Generally, small sample size was acknowledged as a limitation.

Our approach to sample size determination aligns with methodological guidance and recommendations [1] that advise considering the full range of possible quantitative and qualitative methods based on what needs to be learned and the study's available resources:

“Identify the quantitative and qualitative data sources most appropriate for addressing the study’s questions about feasibility and determining whether benchmarks are met. Consider the full range of possible quantitative and qualitative methods and make decisions based on what needs to be learned and the study resources. Keep in mind that even a small sample or data in the form of observation field notes can provide useful information about feasibility.”

Response to Point 3: As for the definition of a clinical trial, we consulted the submission guidelines of PLOS ONE and the relevant links provided for authors. According to the World Health Organization's definition, a clinical trial is any research study that prospectively assigns human participants to health-related interventions to evaluate their effects on health outcomes. Our study, therefore, falls under the definition of a clinical trial.

We are more than willing to engage in revisions to further enhance the quality and rigour of our manuscript. We are committed to ensuring that the final version meets the high standards of PLOS ONE.

Please find the original cover letter below for your reference.

Kind regards,

Barbara Seebacher

---

## [Decision Letter · Decision Letter 1]

18 Mar 2024

PONE-D-23-31858R1Robot-assisted gait training in patients with various neurological diseases: a mixed methods feasibility studyPLOS ONE

Dear Dr. Seebacher,

Thank you for submitting your manuscript to PLOS ONE. After careful consideration, we feel that it has merit but does not fully meet PLOS ONE’s publication criteria as it currently stands. Therefore, we invite you to submit a revised version of the manuscript that addresses the points raised during the review process.

**ACADEMIC EDITOR: Please insert comments here and delete this placeholder text when finished.** Be sure to:Indicate which changes you require for acceptance versus which changes you recommendAddress any conflicts between the reviews so that it's clear which advice the authors should followProvide specific feedback from your evaluation of the manuscriptPlease ensure that your decision is justified on PLOS ONE’s publication criteria and not, for example, on novelty or perceived impact.

We look forward to receiving your revised manuscript.

Kind regards,

Alessandro de Sire, M.D.

Academic Editor

PLOS ONE

Journal Requirements:

"Funder:

Tyromotion GmbH, Graz, Austria

A LEXO Gait Trainer was provided for the Clinic for Rehabilitation Münster to be able to conduct the study. No financial or other donations were given to any personnel involved in the study."

3. In the online submission form, you indicated that The datasets used and/or analysed during the current study are available from the corresponding authors on reasonable request. All data generated or analysed during this study are included in this published article..

4. We note that the original protocol that you have uploaded as a Supporting Information file contains an institutional logo. As this logo is likely copyrighted, we ask that you please remove it from this file and upload an updated version upon resubmission.

5. We note that Figure S1 in your submission contain copyrighted images. All PLOS content is published under the Creative Commons Attribution License (CC BY 4.0), which means that the manuscript, images, and Supporting Information files will be freely available online, and any third party is permitted to access, download, copy, distribute, and use these materials in any way, even commercially, with proper attribution. For more information, see our copyright guidelines: http://journals.plos.org/plosone/s/licenses-and-copyright.

a. You may seek permission from the original copyright holder of Figure S1 to publish the content specifically under the CC BY 4.0 license.

6. We note that Figure S1 includes an image of a participant in the sudy.

7. When completing the data availability statement of the submission form, you indicated that you will make your data available on acceptance. We strongly recommend all authors decide on a data sharing plan before acceptance, as the process can be lengthy and hold up publication timelines. Please note that, though access restrictions are acceptable now, your entire data will need to be made freely accessible if your manuscript is accepted for publication. This policy applies to all data except where public deposition would breach compliance with the protocol approved by your research ethics board. If you are unable to adhere to our open data policy, please kindly revise your statement to explain your reasoning and we will seek the editor's input on an exemption. Please be assured that, once you have provided your new statement, the assessment of your exemption will not hold up the peer review process.

Additional Editor Comments (if provided):

The paper could be accepted after major revisions.

Reviewers' comments:

Reviewer's Responses to Questions

**Comments to the Author**

1. If the authors have adequately addressed your comments raised in a previous round of review and you feel that this manuscript is now acceptable for publication, you may indicate that here to bypass the “Comments to the Author” section, enter your conflict of interest statement in the “Confidential to Editor” section, and submit your "Accept" recommendation.

Reviewer #1: All comments have been addressed

Reviewer #2: (No Response)

2. Is the manuscript technically sound, and do the data support the conclusions?

Reviewer #1: Yes

Reviewer #2: (No Response)

3. Has the statistical analysis been performed appropriately and rigorously? 

Reviewer #1: Yes

Reviewer #2: (No Response)

4. Have the authors made all data underlying the findings in their manuscript fully available?

Reviewer #1: Yes

Reviewer #2: (No Response)

5. Is the manuscript presented in an intelligible fashion and written in standard English?

Reviewer #1: Yes

Reviewer #2: (No Response)

6. Review Comments to the Author

Reviewer #1: Dear Authors,

Thank you for answering to the previous reviwer’s comments. I think you analized a cutting-edge topic, in my opinion the paper is well written.

Reviewer #2: The authors intended to explore the feasibility, acceptability, goal attainment, and preliminary effects of RAGT in patients with common and rare neurological diseases. They recruited 28 inpatients after stroke who will receive the planned intervention. Among 26 patients, RAGT was considered highly feasible. The manuscript was well prepared for the proof of the concept purpose. Only minor comments were listed below.

1. The authors argued no sample size calculation is needed due to the feasibility analysis, which is fine. But also argued “to account for an anticipated attrition rate of 10%”, which somewhat has sample size in mind. Please clarify.

2. Line 253. Effect sizes were calculated as correlation coefficients. Please clarify which two variables’ correlation do you refer to and how to interpret in this content.

3. In the abstract, authors specifically pointed out the count for different diseases but nothing related was discussed later. What will be the main point to talk about in the abstract?

4. As the sample size is extremely small, whether the significant results are replicable is uncertain. So the tone of the results or conclusion should be softened and focus on the feasibility of the study.

7. PLOS authors have the option to publish the peer review history of their article (what does this mean?). If published, this will include your full peer review and any attached files.

Reviewer #1: No

Reviewer #2: No

---

## [Author Response · Author response to Decision Letter 1]

7 Apr 2024

PONE-D-23-31858R1

Robot-assisted gait training in patients with various neurological diseases: a mixed methods feasibility study

PLOS ONE

Dear Dr. de Sire,

Dear Reviewers,

We thank the Editor and Reviewers for their valuable comments and feedback and for giving us a chance to submit a revised manuscript. Please find below our point-by-point responses. The main manuscript and figure S1 caption have been amended accordingly, with any changes highlighted to allow the Editor and Reviewers to check the adaptations made. 

We thank the Editor and Reviewers for their time and effort and hope we have sufficiently responded to the Editor’s and Reviewers’ requests. 

With kind regards,

Barbara Seebacher, on behalf of all the authors

Comments to the Author

Reviewer #1:

Dear Authors,

Thank you for answering to the previous reviewer’s comments. I think you analized a cutting-edge topic, in my opinion the paper is well written.

Dear Reviewer #1, 

Thank you very much for reviewing our manuscript and acknowledging its value.

Reviewer #2:

The authors intended to explore the feasibility, acceptability, goal attainment, and preliminary effects of RAGT in patients with common and rare neurological diseases. They recruited 28 inpatients after stroke who will receive the planned intervention. Among 26 patients, RAGT was considered highly feasible. The manuscript was well prepared for the proof of the concept purpose. Only minor comments were listed below.

Dear Reviewer #2, 

Thank you for reviewing our manuscript, acknowledging its value, and for offering your suggestions to enhance the quality of our manuscript. Please find our point-by-point response as follows.

1. The authors argued no sample size calculation is needed due to the feasibility analysis, which is fine. But also argued “to account for an anticipated attrition rate of 10%”, which somewhat has sample size in mind. Please clarify.

Response:

We appreciate your attention to detail in identifying this inconsistency. Previously, we mentioned that 'Given the study design, methods, and objectives, no extra patients were incorporated to accommodate an expected attrition rate of 10%.' Recognising the lack of value in this statement, we have since deleted it.

2. Line 253. Effect sizes were calculated as correlation coefficients. Please clarify which two variables’ correlation do you refer to and how to interpret in this content.

Response:

Thank you for your valuable feedback. As mentioned, the rank-biserial correlation coefficient (r) is a commonly used effect size measure in the context of the Wilcoxon signed-rank test. This coefficient provides estimates regarding the strength and direction of association between paired samples. To clarify, the paired samples we are referring to in this study are the baseline and post-intervention measures of the respective functional outcomes. 

In response to your suggestion, we have included the following revised statement in the manuscript: ‘Effect sizes were calculated using the rank-biserial correlation coefficient (r) obtained from the Wilcoxon signed-rank test. This coefficient offers insights into the strength and direction of association between paired samples, specifically the baseline and post-intervention measures.’

3. In the abstract, authors specifically pointed out the count for different diseases but nothing related was discussed later. What will be the main point to talk about in the abstract?

Response:

Thank you for bringing this to our attention. Taking your feedback into consideration, we have removed the count for the different diseases from the abstract.

4. As the sample size is extremely small, whether the significant results are replicable is uncertain. So the tone of the results or conclusion should be softened and focus on the feasibility of the study.

Response:

We sincerely value this feedback. Taking into account your suggestion, we have revised our conclusions, which now read as follows: 

‘To our knowledge, there is a gap in the literature regarding the exploration of quantitative RAGT effects alongside patient acceptance and satisfaction using qualitative methods across various neurological conditions. Our study aims to fill this gap by offering valuable insights into understanding patients' needs and identifying necessary adjustments to ensure that RAGT is both enjoyable and maximally effective for patients.

The outcomes of our study imply the feasibility of utilising the LEXO® gait trainer for RAGT among patients with various neurological conditions. This is evidenced by robust recruitment, retention, and adherence rates. Furthermore, the findings underscore the safety of this approach, as indicated by the occurrence of anticipated mild and transient adverse events, alongside its initial effectiveness in enhancing RAGT outcomes among patients with diverse neurological conditions. Moreover, the majority of patients experienced the training positively and expressed satisfaction with their performance and outcomes. While our findings are promising, it is essential to conduct subsequent adequately powered RCTs to validate our results regarding the safety and efficacy of end-effector-based RAGT in patients with MND and SCA. Additionally, further mixed-methods studies are warranted to provide more nuanced insights into patients' perceptions of RAGT with end-effector devices and the influence of the therapeutic relationship on RAGT outcomes.’

---

## [Decision Letter · Decision Letter 2]

5 Jul 2024

Robot-assisted gait training in patients with various neurological diseases: a mixed methods feasibility study

PONE-D-23-31858R2

Dear Dr. Seebacher,

We’re pleased to inform you that your manuscript has been judged scientifically suitable for publication and will be formally accepted for publication once it meets all outstanding technical requirements.

Kind regards,

Domiziano Tarantino, MD

Academic Editor

PLOS ONE

Additional Editor Comments (optional):

Dear Authors,

I am pleased to announce you that the final version of the manuscript has been accepted for publication.

Best regards

Reviewers' comments:

Reviewer's Responses to Questions

**Comments to the Author**

1. If the authors have adequately addressed your comments raised in a previous round of review and you feel that this manuscript is now acceptable for publication, you may indicate that here to bypass the “Comments to the Author” section, enter your conflict of interest statement in the “Confidential to Editor” section, and submit your "Accept" recommendation.

Reviewer #2: All comments have been addressed

2. Is the manuscript technically sound, and do the data support the conclusions?

Reviewer #2: (No Response)

3. Has the statistical analysis been performed appropriately and rigorously? 

Reviewer #2: (No Response)

4. Have the authors made all data underlying the findings in their manuscript fully available?

Reviewer #2: (No Response)

5. Is the manuscript presented in an intelligible fashion and written in standard English?

Reviewer #2: (No Response)

6. Review Comments to the Author

Reviewer #2: All the raised comments have been successfully addressed. This reviewer has no further comments on this manuscript.

7. PLOS authors have the option to publish the peer review history of their article (what does this mean?). If published, this will include your full peer review and any attached files.

Reviewer #2: No

---

## [Editor Report · Acceptance letter]

15 Jul 2024

PONE-D-23-31858R2 

PLOS ONE

Dear Dr. Seebacher, 

I'm pleased to inform you that your manuscript has been deemed suitable for publication in PLOS ONE. Congratulations! Your manuscript is now being handed over to our production team.

Kind regards, 

on behalf of

Dr. Domiziano Tarantino 

Academic Editor

PLOS ONE